# An analysis of global legislation and regulation related to drowning prevention

Ryan Essex●[1,2]ᵒ*, Jagnoor Jagnoor[1,2]ᵒ

1 The George Institute for Global Health, Sydney, Australia, 2 University of New South Wales, Sydney, Australia

ᵒ All authors contributed equally to this work.
* ressex@georgeinstitute.org.au

## Abstract

Drowning remains a leading but neglected cause of injury-related death worldwide. While some forms of regulation, such as pool fencing, have demonstrated clear effectiveness, less is known about the broader influence of legislation and regulatory measures on drowning mortality. This study provides the first global analysis of drowning-prevention legislative coverage, utilising secondary analysis of cross-sectional data, to explore country-level associations between legislation/regulation and drowning rates. Utilising the World Health Organization (WHO) Global and Regional Status Reports on Drowning, alongside economic, environmental, governance, and health-system indicators, we assembled a cross-section of 127 countries. We estimated several models that included the full set of national legislation indicators and nine covariates, complemented by a random-forest analysis. Across specifications, we observed no consistent association at the country level between the presence or count of legal measures and lower drowning mortality. By contrast, structural determinants were repeatedly implicated. For example, in our main models, higher per-capita alcohol use was positively associated with drowning mortality (IRR = 1.18, 95% CI [0.99, 1.40]), while better access to safe water/sanitation was protective (IRR = 0.84, 95% CI [0.71, 0.99]). Exploratory interactions examining the influence of institutional/governance capacity suggested that public-pool fencing and, more tentatively, alcohol restrictions near water, were more protective where institutional capacity was stronger. Our findings support the idea that legalisation/regulation is likely to be most effective when embedded within broader investments in infrastructure, service delivery, and enforcement capacity.

## Introduction

When it comes to drowning prevention, throughout the literature we can find several examples where regulation and legislation have been effective in reducing drowning mortality and associated risks. Take the example of pool fencing legislation in

---

**Data availability statement:** All data utilised in this study is publicly available and cited in the main text. All data can be found at the following URLs: WHO Global Status Report on Drowning: https://www.who.int/teams/social-determinants-of-health/safety-and-mobility/global-report-on-drowning-prevention WHO Western Pacific Regional Status Report on Drowning: https://www.who.int/publications/i/item/9789290619581 WHO Southeast Asia Regional Status Reports on Drowning: https://www.who.int/publications/i/item/9789290221098 Varieties of Democracy (V-DEM): https://www.v-dem.net/ World Justice Project (WJP): https://worldjusticeproject.org/ Global Health Security (GHS): https://ghsindex.org/ World Banks World Data Indicators (WDI): https://databank.worldbank.org/source/world-development-indicators.

**Funding:** The authors received no specific funding for this work.

**Competing interests:** The authors have declared that no competing interests exist.

Australia. Major steps toward fencing private pools in Australia started to take shape in the 1990's, two decades after childhood drowning in backyard pools was identified as an issue [1]. While this took some time to get right and while implementation varied state by state, these laws today have proven effective, with drowning deaths dropping by about 50% in children aged 0–4 years with pool fencing playing a substantial role in this decrease [2]. Elsewhere, regulation was found to be equally as effective when it came to personal flotation devices. In another study from Australia, it was found that the introduction of legislation that mandated the use of personal flotation devices amongst recreational boaters decreased drowning fatalities by 70% when looking at mortality in the periods of 5 years pre-post intervention [3]. Internationally, we find regulatory and policy efforts for those working on or around water with extensive international regulation in relation to vessel design and safety for example [4]. At the same time however, we know little about many forms of regulation and how they directly and indirectly impact drowning prevention efforts, with few studies having focused directly on the impact of legislation and regulation in this space. There are a small number of exceptions here when it comes to drowning prevention, with a recent scoping review identifying 25 papers that provided some insights into the effectiveness of legislation for a range of outcomes, from mandating lifejackets to regulating recreational space in open water [5]. One of the findings of this study was that while legislative and regulatory efforts appeared to be promising in curbing drowning deaths, their success hinged on a number of factors, like compliance, enforcement and having the appropriate expertise when it came to development and implementation. While generally, the studies in this review were often small and relied on case studies, amongst these papers were a small number of larger studies that examined the effect of legislation and regulation on a broader scale. For example, Quan, Mills [6] examined regulation across 30 United States (US) states related to open water safety. This study found that states with greater regulation generally had lower open water drowning death rates. Amongst states without any form of regulation, drowning rates were three and four times higher amongst young people and non-white residents respectively, compared to those who had all five types of regulation.

As a whole, the literature suggests that regulation, under the right conditions, can be a powerful tool to curb drowning deaths. There are still however shortcomings and gaps. We know little about the influence of regulation on drowning in certain sectors for example. The evidence that we have is also generally based on smaller and sector-specific studies, with most studies focused on discreet pieces of legislation, like those focused on pool fencing or personal flotation devices. From these studies we can begin to put together some of the causal pathways that make regulation effective, from altering the environment, to changing behaviours and norms, however, there also remains a paucity of evidence here.

One further area that has been overlooked in this space has been the global landscape of legislative and regulatory approaches related to drowning prevention. The recent World Health Organization (WHO) Global Status Report on Drowning Prevention [7], along with two earlier regional reports [8,9] provide valuable resources in this

respect, in that they give some insight into the broader regulatory and legislative measures implemented by countries to curb drowning, providing data on the coverage and type of legislation/regulation implemented by countries. Using this data this study sought to add to the modest literature above, providing the first global examination of regulatory and legislative measures related to drowning prevention.

### Research aims

This study therefore sought to examine 1) the extent to which legislative measures, both individually and in aggregate, were associated with drowning mortality rates 2) the specific combinations of legal, environmental, health and governance factors that predicted countries with differential drowning burdens and 3) whether strength of enforcement and institutional capacity were associated with the effectiveness of legislation/regulation. Our analysis is ecological and cross-sectional. Consequently, estimates may reflect endogeneity (e.g., countries adopting laws in response to high drowning) and measurement error; we therefore advise caution and treat findings as descriptive of country-level patterns.

### Methods

We addressed the above research aims through three complementary approaches. Aims 1–2 used negative binomial regression to estimate associations between individual laws (Model M1), aggregate legislative counts (Model M2), and drowning mortality, adjusted for structural confounders including economic development, health systems, and governance. Aim 2 was further addressed using Random Forest classification to identify optimal combinations of legal environmental, and institutional predictors of drowning burden. Aim 3 employed interaction models testing whether governance capacity moderated legislative effectiveness, supplemented by regional models (M3–M5) directly measuring enforcement implementation strength.

### Data sources

Data was assembled at country level for 2021, with variables collated from a range of sources. Each data source and the variables included in this analysis are included in S1 Table. A brief description is provided here. The core outcomes and explanatory variables for this study come from the Global Status Report on Drowning report [7] and the Western Pacific and South East Asian Regional Status Reports on Drowning [8,9]. Drowning rates per 100,000 population were extracted from the global status report (and reconstructed as count data; see below), along with the presence of a national drowning prevention plan, and the presence or absence of a national drowning prevention strategy and the six legislative/regulatory measures listed, including fencing requirements for private and public pools, domestic water-transport safety regulations, life-jacket mandates, and alcohol restrictions near waterbodies. We extracted all legislative measures and the extent to which they were enforced from the regional status reports. We also included covariates from a range of sources. Variables included a range of economic, legal, environmental, geographic and health-system related metrics that all had potential bearing on rates of drowning and the implementation and enforcement of legislation/regulation. These were extracted from the Varieties of Democracy (V-DEM) [10], World Justice Project (WJP) [ 11], Global Health Security (GHS) [12], and the World Banks World Data Indicators (WDI) [13] datasets. We utilised data from 2021 in all cases, except for the WJP where we used the 2023 rule of law index. For average temperature we utilised mean temperature form 1995–2020.

### Data transformation

We derived a number of variables from the global and regional status reports.

From the global status reports, we extracted data on seven distinct legislative/regulatory domains: (1) national drowning prevention strategy, (2) disaster risk management policy specifying drowning, (3) fencing requirements for private pools, (4) fencing requirements for public pools, (5) domestic water transport safety requirements, (6) lifejacket use mandates, and (7)

alcohol regulation near waterbodies. Legislative/regulatory presence was originally categorised as national/sub-national/no; we recoded this as a binary (yes/no) variable to indicate presence (at any jurisdictional level) or absence in each country. We also computed a total legislative count by summing across six of these domains (excluding the national drowning prevention strategy, which we modelled separately as it represents strategic coordination rather than specific regulatory mechanisms). This provided a total legislative measures variable scored 0-6, where higher scores indicate more comprehensive legislative coverage across key drowning prevention domains. As the regional reports included more granular data on legislative/regulatory mechanisms and the extent of their enforcement (which took the extent to which each law was enforced, rated 0-10), we opted to calculate two variables, this included the total number of legislative instruments reported (with a possible range 0-19) and a total enforcement score (with a possible range of 0-190). Specifically, the regional reports contain data across 23 specific legislative/regulatory mechanisms organized into five domains: (1) maritime safety, (2) child safety, (3) water safety competencies, (4) disaster risk reduction, (5) drowning surveillance. Total legislative count (range 0-19) was summed by observing a simple count of how many of the 19 legislative/regulatory mechanisms were reported as present in each country or territory, regardless of enforcement strength. This provided a measure of legislative breadth. Total enforcement score (range 0-190) took implementation scores on a 0-10 scale (with lower score indicating less enforcement/implementation). We summed these ratings across all 19 items to create an overall enforcement strength score. This measure captured not just whether legislation exists, but the extent to which it was actively implemented and enforced. These two measures (legislative count and enforcement score) provided complementary information, notably, a country might have many laws on the books (high count) but weak implementation (low enforcement score), or conversely, fewer laws that are strongly enforced. We examined both independently (Models M3-M4) and their interaction (Model M5) to test whether enforcement capacity moderates the association between legislative presence and drowning mortality. To ensure consistency across all variables, exposure to natural disasters (from the GHS data see below) was reverse scored, as in the original dataset lower scores meant greater exposure to natural disasters. To aid analysis where we utilised count models, we worked with country counts of drowning deaths and a population offset. As only rates were available, deaths were reconstructed (rate × population / 100,000) and rounded to the nearest integer. This rounding introduced minor approximation error, though given the scale of the counts involved, the effect on model estimates is negligible. All continuous covariates were z scored before modelling.

## Variables

### Outcome

The major outcome of interest in this study was drowning mortality per 100,000 population as reported in the Global Status report [7].

### Key explanatory variables/policy exposures

The main outcome variables were each of the legislative/regulatory measures from the global status report; these included fencing requirements for private and public pools, domestic water-transport safety regulations, life-jacket mandates, and alcohol restrictions near waterbodies. We also included whether the country had national water safety plan. A derived variable was also included that summed the number of legislative instruments (including a national water safety plan). From the regional reports, two variables were included, the number of legislative instruments reported in each country and an index that represented the total strength of their enforcement. In total this provided eight explanatory variables that were included in the main models, with regional models included as supplementary material.

### Covariate selection

Alongside the above outcomes and to control for confounding we assembled 24 candidate controls spanning climate and geography (average temperature, precipitation), demographics (child population share, urbanisation, tourist visits),

health-system capacity (hospital beds, physicians, per-capita health spending), risk exposures (alcohol use, natural-disaster and environmental hazards), and institutional/governance quality (rule of law, regulatory enforcement, government effectiveness). To inform final covariate selection we applied elastic-net stability selection to identify the most reliable confounders. Across 500 bootstrap resamples, an elastic-net model with ten-fold cross-validation was refit, and inclusion frequency (the proportion of resamples in which its coefficient was non-zero) was recorded. The final covariates that were selected were gross domestic product at purchasing power parity (GDP PPP), average surface temperature, legislative transparency and enforcement (V-DEM), per-capita alcohol consumption (WDI), health-sector capacity, urbanisation, natural-disaster exposure, access to water and sanitation, and public-health spending (GHS). The selection of these variables was driven by the above model and existing theoretical and empirical evidence on drowning prevention [14]. Each of these variables, a description and their source is included in S1 Table.

## Causal considerations and covariate classification

To formalise assumptions about confounding, we constructed a directed acyclic graph (DAG) representing hypothesized causal relationships between legislation, structural factors, and drowning mortality (S2 Fig). The DAG identified multiple sufficient adjustment sets; all included economic development and governance capacity as essential confounders. Our final control set (nine covariates) represents a sufficient adjustment set under the DAG assumptions. We classified urbanization and public health spending as potential partial mediators (economic development may influence these, which in turn affect both legislation and drowning), though both also exhibit strong confounding properties. Sensitivity analyses excluding these variables are reported in the supplement. No collider bias was identified under the DAG structure.

## Analysis

### Main models

To estimate the association between legislation/regulation and drowning mortality, we modelled reconstructed death counts (rate × population/ 100,000) using negative binomial regression (NB2) with a log link function and log(population/100,000) offset to accommodate over-dispersion in count data. All models used robust standard errors and reported incidence rate ratios (IRRs) with 95% CIs. We estimate three nested specifications that differ only in how legal coverage was represented; each includes the common control set (see above):

- M0 — controls only: establishing a baseline relationship between the controls and drowning deaths.

  - M1 — controls + eight policy indicators (from the global status report): asking which specific laws matter the most. To avoid quasi-separation in small cells, indicators that were non-varying or ultra-rare in the estimation sample were excluded.

- M2 — controls + number of legislative measures (from the global status report): testing whether having more laws in total was linked to lower drowning rates.

Regional variables (total laws, enforcement score, and their interaction) were available for a much smaller country subset (n = 19). This made estimates unstable. To prevent over-fitting and avoid conflating samples, we report the regional models (M3–M5) as supplementary material (see below).

### Robustness and diagnostics

We subjected the main specifications to several robustness exercises designed to probe sensitivity to estimator choice, influential observations, and collinearity among the structural controls. First, we re-estimated the negative binomial models with a population offset using a Poisson pseudo-maximum likelihood (PPML) estimator with a log link, holding the sample fixed across estimators so that any differences reflected modelling rather than composition. Second, to assess the influence

of outliers, we implemented an influence-trim procedure: for each model we fit a Poisson start, computed Cook's distances, removed the top five percent of observations, and re-fit the corresponding NB2 specification on the trimmed sample. To address potential over-parameterisation and shared variance among the controls, we implemented a dimension-reduction exercise. Specifically, we performed principal components analysis (PCA) on the structural controls and replaced the original controls with the leading components required to explain approximately 80% of their variance. We then re-estimated the policy model using NB2 (with Poisson fallback where necessary), retaining the policy indicators unchanged.

We evaluated the stability of the three main NB2 specifications using diagnostics and robustness analyses tailored to count outcomes with a population offset. For each model we simulated residuals and examined tests of dispersion, zero inflation, and uniformity; we assessed multicollinearity using variance inflation factors and we tested spatial autocorrelation using Moran's I based on k-nearest-neighbour weights derived from country centroids.

### Exploratory machine-learning classification of drowning risk

To complement parametric models, we trained a global random-forest regression model using drowning mortality rate as the outcome. Predictors comprised the full set of national legislation indicators alongside the nine covariates. We used 1,000 trees with out-of-bag prediction and computed permutation importance for each feature. To quantify uncertainty in importances, we bootstrapped the rows (B ≈ 300 resamples), re-fit the forest in each resample, and summarised the distribution of importances to obtain means and percentile 95% CIs. This non-parametric approach provided a complementary ranking of legal and contextual factors associated with variation in drowning risk, without imposing linearity or additivity. It is worth noting that permutation importance reflects the predictive contribution of a given variable to the model, that is, how much model accuracy deteriorates when that variable is shuffled. It does not imply a causal relationship between the variable and the outcome.

### Interaction between governance capacity and legislative measures

To examine whether the influence of each legislative/regulatory measure varied with levels of enforcement, we considered four metrics: V-DEM's legislative transparency/enforcement and respect for rule-of-law indices, and the World Justice Project's rule-of-law and regulatory enforcement indices. While similar, each metric measures a slightly different element of enforcement and rule of law. V-DEM's legislative transparency/enforcement index captures the extent to which laws are made in a transparent way and whether legislation is actually implemented and enforced in practice, while the respect for rule of law index captures whether legislation is applied impartially and in a fair and transparent way. World Justice Project's rule-of-law index reflects how well a country adheres to the rule of law across eight dimensions, including constraints on government powers, absence of corruption and fundamental rights, while the regulatory enforcement index captures how effectively and fairly regulations are implemented and enforced in practice.

For each index, we fitted a negative binomial (NB2) count model with a population offset, including the seven global legislative indicators (presence/absence of each measure, including a national drowning-prevention strategy) and the same structural covariates as in the main models. Enforcement capacity was standardised (mean 0, SD 1) and entered both as main effects and in multiplicative terms with every law (e.g., lifejacket requirement × regulatory enforcement), allowing the model to estimate the baseline association of capacity and the incremental change in a law's incidence rate ratio (IRR) as enforcement capacity increased. To aid interpretation, we derived predicted IRRs for countries with and without the law in question at –1 SD, the mean, and +1 SD of capacity,

## Results

### Main models

We estimated associations between drowning prevention legislation and mortality using negative binomial regression, adjusting for nine structural confounders including economic development, governance capacity, health systems, and

environmental factors (M0-M2; N = 104–127). Confounder coefficients are reported in Table 1 for transparency but are not interpreted as causal effects. When individual laws were examined (M1, N = 104), none of the seven legislative indicators reached conventional significance; point estimates fluctuated around the null, including for private-pool fencing (IRR = 0.64, 95% CI: 0.39–1.05, p = 0.17) and water-transport safety (IRR = 0.76, 95% CI: 0.52–1.10, p = 0.15). A simple count of national legislative measures (M2, N = 127) showed no association with mortality (IRR = 1.03, 95% CI: 0.97–1.10, p = 0.35). Overall, after adjusting for structural confounders, we found no systematic evidence that the presence of specific national laws, or having more laws in total, was associated with lower drowning mortality. Results are summarized in Fig 1 and Table 1.

Because the regional dataset covered 19 countries, estimates were imprecise: coefficients on total regional laws, enforcement strength, and their interaction clustered tightly around the null with very wide confidence intervals and no consistent pattern across models. Given the small N and corresponding uncertainty, we treat these as supplementary and interpret them with caution (S1 Fig and S5 Table).

## Robustness and diagnostics

Results were generally stable across alternative estimators and sample perturbations. Re-estimating the three specifications with Poisson pseudo–maximum likelihood on a fixed common sample did not overturn the main findings. In NB2, alcohol remained positively associated with drowning mortality (M0 IRR = 1.236, p < .05; M2 IRR = 1.240, p < .05) and access to safe water/sanitation remained protective (M0 IRR = 0.809, p < .05; M2 IRR = 0.811, p < .05). Under PPML these effects were attenuated and imprecise, and, consistent with the main text, neither the individual laws nor the total count of laws were systematically associated with mortality (PPML M2 total-laws IRR = 1.198, n.s.). A single PPML coefficient on "Disaster policy" was positive (IRR = 2.038, p < .05) but was not reproduced in NB2 and should be read cautiously given the smaller M1 sample (N = 71) and multiple comparisons.

An influence-trimmed analysis, dropping the top 5% of cases by Cook's D and re-fitting NB2 on the trimmed rows yielded very similar patterns. Alcohol use was again associated with increased drowning mortality (M0 IRR = 1.298, p < .01; M2 IRR = 1.297, p < .01) and water/sanitation remained protective (M0 IRR = 0.743, p < .05; M2 IRR = 0.747, p < .05). Urbanisation became more precisely positive (M0 IRR = 1.335, p < .01; M2 IRR = 1.357, p < .01), and public-health spending was clearly protective across trimmed models (IRR ≈ 0.70, p < .001). To address shared variance among controls, we replaced them with principal components explaining ~80% of their variance and re-estimated the policy model. PC1–PC3 were strongly protective (IRR = 0.778, 0.829, 0.857; all p ≤ .05), while policy indicators again clustered around the null, with at most a marginal effect for private-pool fencing (IRR = 0.590, p < .10). Finally, diagnostics indicated no major misspecification: dispersion, zero-inflation, and uniformity tests were acceptable (p = 0.377–1.000), spatial autocorrelation in residuals was negligible (Moran's I ≈ −0.03, p ≈ 0.63–0.65), and collinearity was moderate (max VIF ≈ 8.4 in M0/M2; 12.4 in M1) and consistent with the larger policy block. All of these results are included as supplementary material (S6-S10 Tables).

We constructed a directed acyclic graph (DAG) formalizing assumed causal relationships (S1 Fig). The DAG identified our nine-covariate set as a sufficient adjustment set. Urbanization and public health spending were classified as potential partial mediators but retained as confounders based on strong confounding properties; sensitivity analyses excluding them are reported in supplements (S12 Table)

## Machine-learning classification of drowning risk

To complement the parametric models, we fitted a global random-forest regression to the log drowning-mortality rate, using the full set of national legislation indicators alongside the nine structural covariates. The ranking was dominated by structural conditions. Public-health spending, access to safe water and sanitation, GDP, and urbanisation showed the largest and most precise importances (mean 0.269 [0.167, 0.422]; 0.245 [0.155, 0.366]; 0.237 [0.144, 0.361]; 0.226 [0.111,

**Table 1. Main NB2 models (IRR and p values).**

| | M0 | M2 | M1 |
|---|---|---|---|
| GDP | 0.911 | 0.900 | 0.901 |
| | (0.136) | (0.141) | (0.183) |
| Avg temp | 1.092 | 1.079 | 1.094 |
| | (0.084) | (0.079) | (0.085) |
| Legislative enforcement | 0.950 | 0.958 | 0.970 |
| | (0.075) | (0.076) | (0.100) |
| Alcohol | 1.178* | 1.176* | 1.186 |
| | (0.076) | (0.076) | (0.131) |
| Health-sector robustness | 0.992 | 0.971 | 0.978 |
| | (0.102) | (0.097) | (0.109) |
| Urbanisation | 1.231+ | 1.235+ | 1.179 |
| | (0.134) | (0.136) | (0.142) |
| Disaster exposure | 1.120 | 1.124 | 1.120 |
| | (0.089) | (0.090) | (0.098) |
| Water & sanitation | 0.838* | 0.835* | 0.805+ |
| | (0.073) | (0.072) | (0.093) |
| Public-health spend | 0.847 | 0.851 | 0.882 |
| | (0.120) | (0.121) | (0.180) |
| National strategy | | | 1.044 |
| | | | (0.234) |
| Disaster policy | | | 1.104 |
| | | | (0.276) |
| Private-pool fencing | | | 0.638 |
| | | | (0.199) |
| Public-pool fencing | | | 1.039 |
| | | | (0.282) |
| Water-transport safety | | | 0.759 |
| | | | (0.156) |
| Lifejacket requirement | | | 1.279 |
| | | | (0.272) |
| Alcohol regulation near water | | | 1.163 |
| | | | (0.182) |
| Total laws (global) | | 1.032 | |
| | | (0.035) | |
| Num.Obs. | 127 | 127 | 104 |
| RMSE | 2454.75 | 2433.96 | 2490.38 |

Note: + p < 0.1, * p < 0.05, ** p < 0.01, *** p < 0.001

Note: confounder coefficients are presented for completeness but not interpreted as causal effects.

0.354], respectively). By contrast, the national legislation indicators exhibited low importances with intervals close to zero. Taken together, this non-parametric exercise aligns with the regression results with structural determinants accounting for most of the cross-national signal in drowning mortality, while the presence/absence of individual laws contributed comparatively little predictive power at the national level. Random forest results are summarised below in Fig 2 and as supplementary material (S11 Table).

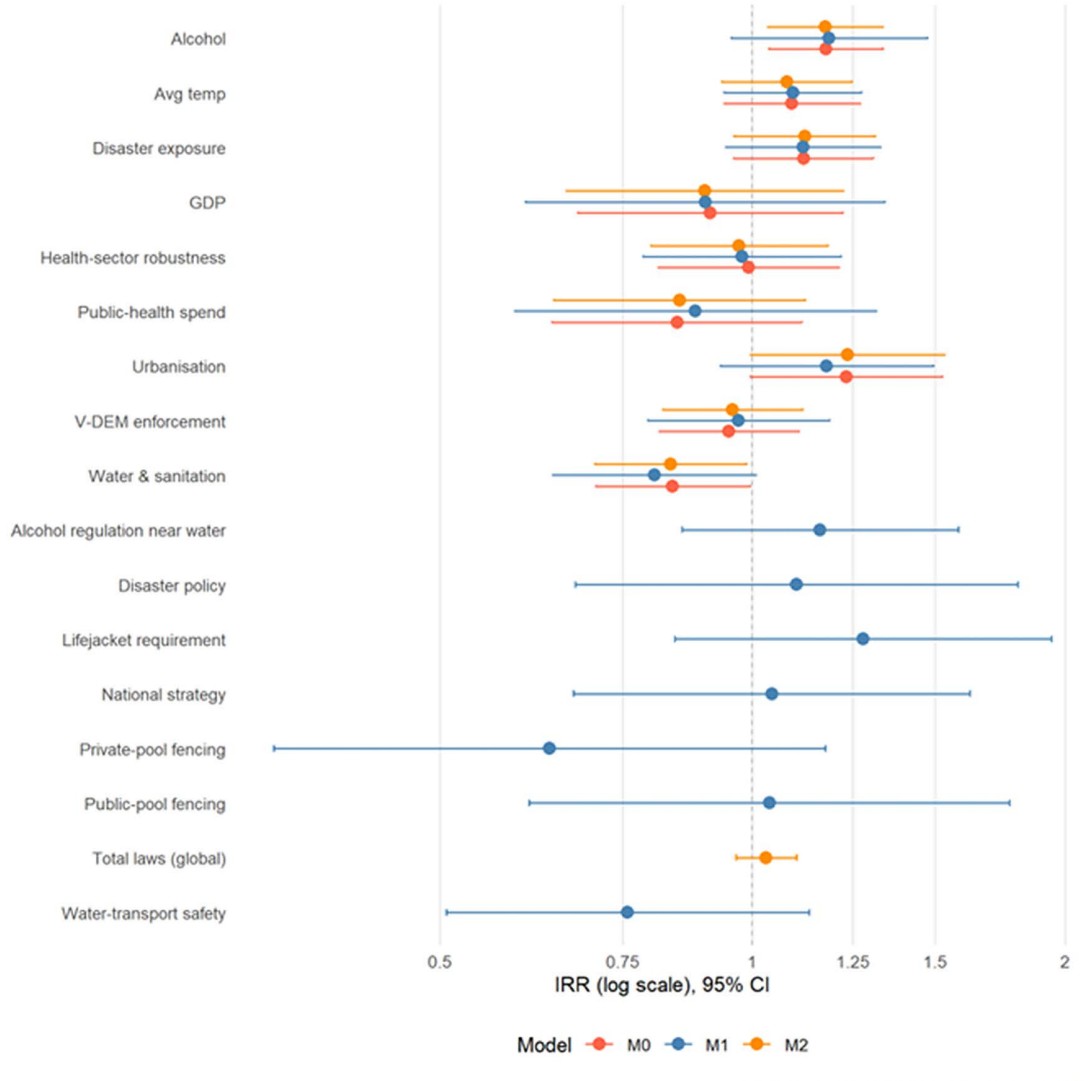

**Fig 1. Forest plot of main NB2 models (IRR and p values).**

## Interaction between governance capacity and legislative measures

We next asked whether drowning mortality was associated with country institutional/governance capacity. To examine this, we utilised four governance proxies (V-DEM legislative transparency/enforcement; V-DEM respect for rule of law; WJP regulatory enforcement; WJP rule of law). We estimated NB2 models including the law main effects and all law × capacity interactions. Across measures, interaction estimates were generally small and imprecise, clustering near zero (Fig 3 and S12 Table). There were two exceptions. First, public-pool fencing showed a negative interaction with V-DEM respect for rule of law (β = −0.859, SE = 0.329, p = 0.009), consistent with a stronger protective association where governance capacity was higher. Second, alcohol restrictions near water displayed negative interactions with WJP regulatory enforcement (β = −0.556, SE = 0.335, p = 0.097) and with V-DEM respect for rule of law (β = −0.451, SE = 0.243, p = 0.063), however did not reach significance. For disaster policy, water-transport safety, lifejacket requirements, and private-pool fencing,

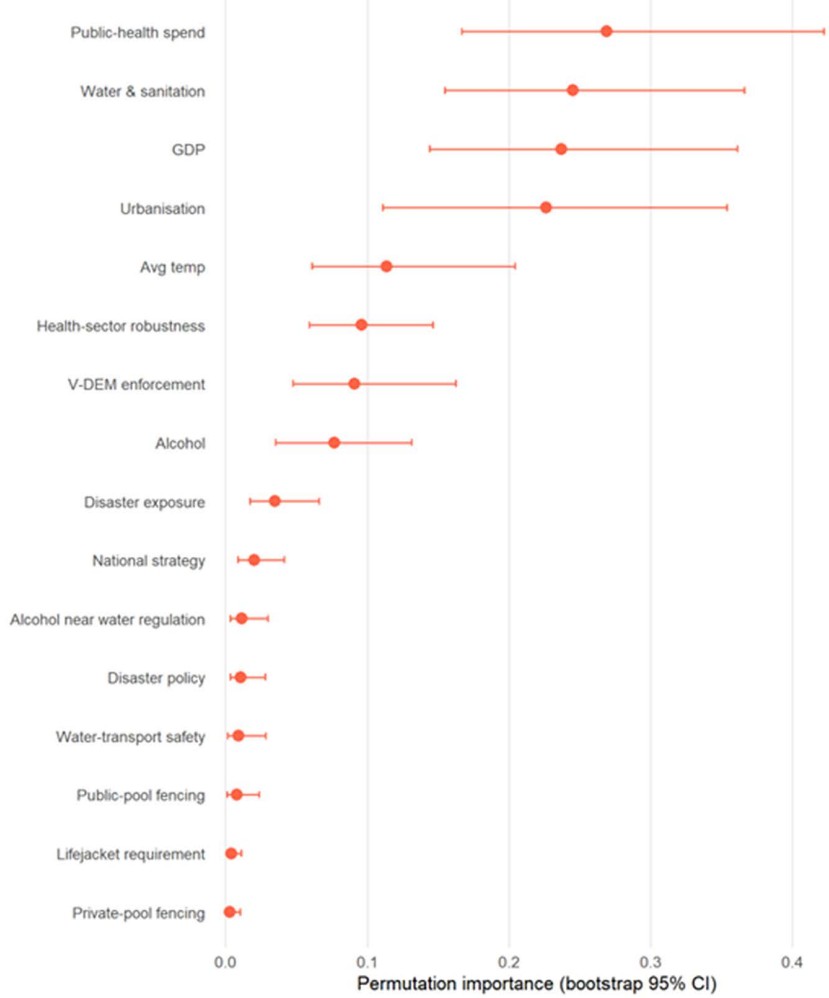

**Fig 2. Random Forest: Importance of legislation and other factors in relation to countries with lower v higher rates of drowning (with bootstrap CIs).**

interaction terms were close to null across all capacity proxies (all p > 0.10). The aggregate "total laws" count also showed no systematic interaction with capacity (Table 2).

## Discussion

This study set out to test whether national legislative and regulatory measures, individually and in aggregate, were associated with drowning mortality; to see which combinations of legal, environmental, health-system and governance factors best separated higher- from lower-burden settings; and to assess whether any legal influence depended on institutional capacity and enforcement. Taken together, the results point in a consistent direction. After accounting for structural covariates, we did not detect systematic associations between the presence or number of national laws and lower drowning mortality at the ecological, country level. Instead, variation in drowning mortality was most strongly linked to broader conditions, access to safe water and sanitation, alcohol consumption, urbanisation, and related socio-economic factors. Where we did find associations, they were conditional on country governance and institutional capacity. That is,

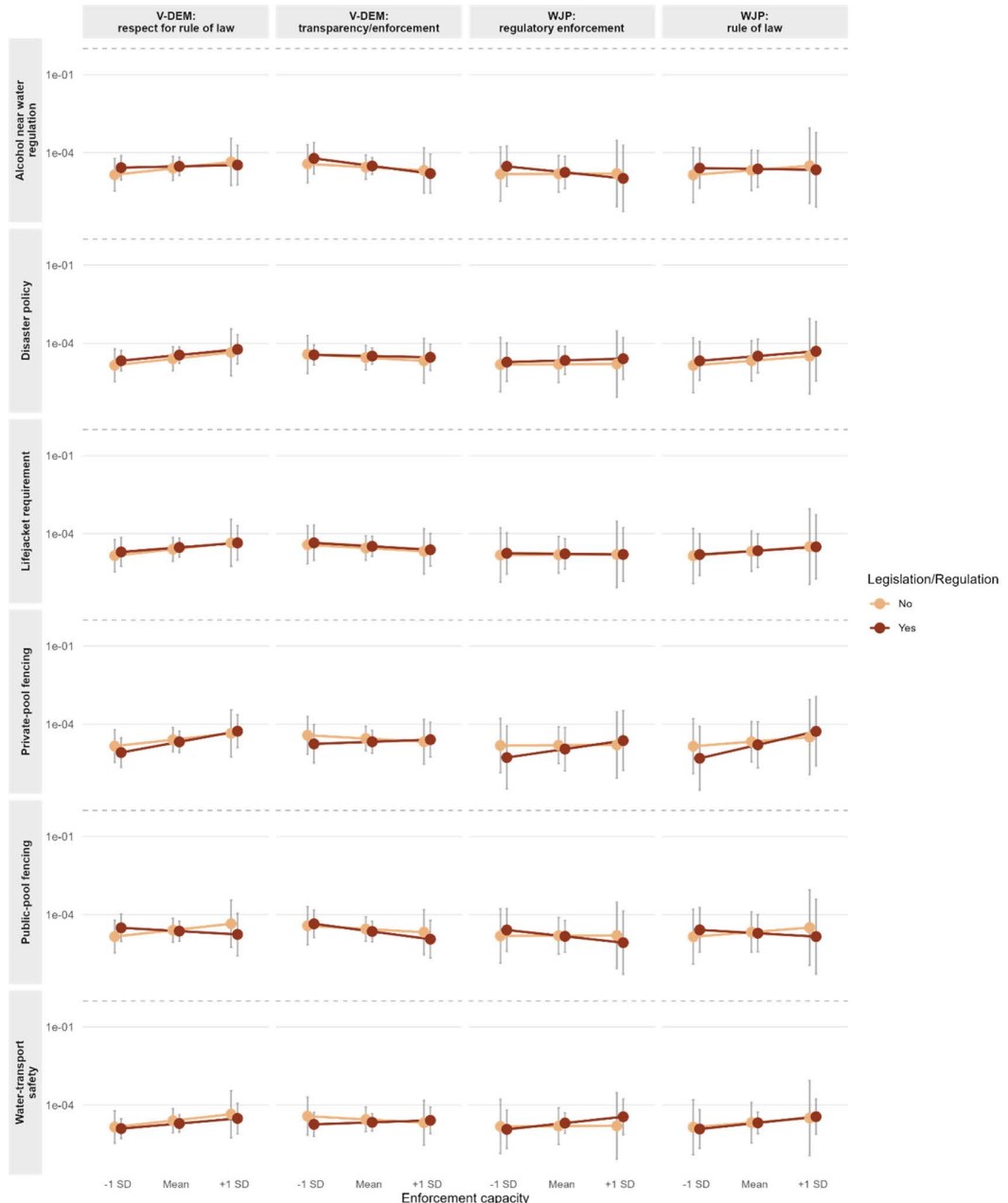

**Fig 3. Interaction between enforcement capacity × law.**

public-pool fencing and, more tentatively, alcohol restrictions near water were more protective where rule-of-law and regulatory enforcement were stronger. This aligns with case studies showing effectiveness in targeted contexts while suggesting effects do not aggregate cleanly at the country level, likely because implementation, coverage, and compliance vary. As a whole, our global cross-section shows that legislative coverage on its own was not reliably associated with lower drowning related mortality; any benefits appear to rely on the institutional and infrastructural context in which those laws were introduced and implemented.

**Table 2. Enforcement capacity × law interaction coefficients.**

| | Est | SE | p | Est | SE | p | Est | SE | p | Est | SE | p |
|---|---|---|---|---|---|---|---|---|---|---|---|---|
| Law | V-DEM: transparency/ enforcement | | | WJP: regulatory enforcement | | | WJP: rule of law | | | V-DEM: respect for rule of law | | |
| Disaster policy | 0.181 | 0.481 | 0.708 | 0.126 | 0.575 | 0.827 | 0.018 | 0.546 | 0.973 | -0.064 | 0.451 | 0.886 |
| Private-pool fencing | 0.475 | 0.507 | 0.349 | 0.723 | 0.716 | 0.313 | 0.787 | 0.691 | 0.255 | 0.376 | 0.553 | 0.497 |
| Public-pool fencing | -0.392 | 0.298 | 0.189 | -0.585 | 0.596 | 0.326 | -0.701 | 0.549 | 0.202 | -0.859 | 0.329 | 0.009 |
| Water-transport safety | 0.463 | 0.455 | 0.310 | 0.530 | 0.694 | 0.446 | 0.142 | 0.826 | 0.864 | -0.117 | 0.419 | 0.780 |
| Lifejacket requirement | -0.019 | 0.273 | 0.946 | -0.077 | 0.356 | 0.828 | -0.064 | 0.380 | 0.867 | -0.166 | 0.273 | 0.544 |
| Alcohol near water regulation | -0.384 | 0.242 | 0.112 | -0.556 | 0.335 | 0.097 | -0.481 | 0.308 | 0.119 | -0.451 | 0.243 | 0.063 |
| Total laws | -0.031 | 0.242 | 0.899 | 0.016 | 0.311 | 0.958 | 0.016 | 0.297 | 0.958 | 0.145 | 0.243 | 0.552 |

We treat these findings with some degree of caution, given the broader literature and the limitation of this study (see below). That is, while there have been no studies to examine global legislative coverage by country, our findings stand in contrast to much of the literature, that speaks to the effectiveness of regulation and policy in relation to drowning whether this be in relation to pool fencing [15] or the regulation of personal flotation devices [16]. Notably, our results stand in contrast to other studies that have employed similarly broad designs, at least when looking at legislation at a national level [6]. It is important to be clear about what our null findings mean. The absence of a statistically significant association at the ecological, country level does not mean that laws are ineffective or policy-irrelevant. National-level aggregation masks the considerable heterogeneity in how laws are designed, implemented, and enforced across settings. These null results should therefore be read with caution, as a reflection of the limits of ecological analysis, not as evidence against legislation as a means to curb drowning. Put another way, while this study found no evidence of an association at an ecological level, it does not follow that legislation is ineffective. Our other finding helps explain this and also better aligns with the literature, that is, our finding also suggest that the effectiveness of legislation was dependent on a range of factors like governance and institutional capacity, including the implementation and enforcement of legislation [5]. These findings are important in other ways, beyond the implications for understanding the role of legislation in drowning prevention. The results also speak to the need for better data in this space, not only related to drowning mortality, but in relation to legislation/regulation and its effectiveness. This echoes the findings of a recent scoping review that, amongst other things, reached similar conclusions [5], notably around the need for greater data, not only in relation to legislative and regulatory coverage, but in relation to the qualitative elements of the regulation or legislation in question. Finally, and one thing that this analysis points out is that broader social, political and environmental factors are also influential when it comes to drowning prevention, consistent with much of the previous literature in this space [17,18]. Taken with our findings that enforcement capacity may have differential influence on various legislative measures, this directs us to the question of what (legislation/regulation) works, for whom in what circumstance. These results suggest it is important we think about legislation alongside conditions that shape drowning risk. Because structural determinants, safe water access and sanitation, alcohol use, urbanisation, and health-system capacity were found to be most influential, legal reforms are likely to be most effective when integrated alongside these considerations. For example, alcohol restrictions near water may be most effective alongside alcohol-harm reduction more generally; pool-fencing is likely going to be most effective when backed by subsidies, inspections and enforcement; transport-safety rules should be matched with training and coastguard resourcing and so on. To illustrate, pool fencing requirements have demonstrated clear effectiveness in high-income settings like Australia, where compliance is supported by inspections and public awareness. In lower-capacity settings, however, the same legislation may exist on paper without the institutional infrastructure needed to enforce it, meaning coverage and compliance remain low. Similarly, alcohol restrictions near water may reduce risk in settings with established enforcement mechanisms, but where regulatory oversight is limited, such provisions may have little practical effect. These

examples, if correct, suggest that the effectiveness of legislation is not inherent to the law itself, but to the conditions in which it operates.

## Limitations

At least four major limitations are worth discussing. First, the study is ecological and cross-sectional. We analysed country-level aggregates at a single point in time, so any associations we observed were averages over heterogeneous populations. The ecological fallacy applies directly here: a null or weak association at the national level does not imply that laws fail for specific groups or settings. Relatedly, we cannot establish temporal ordering. Laws may have been introduced because drowning was high, and without panel data we cannot separate reactive legislation from causal effects on later mortality. Second, measurement is coarse. The global series records the presence of legal measures but not their content, coverage, implementation, or compliance. Laws can vary widely in scope and enforceability; we lack systematic information on sub-national rules, enforcement effort, and sanctions. Our outcome is likewise an aggregate rate without age- or sex-specific detail, which limits our ability to detect targeted policy effects. Relatedly, national-level aggregation may obscure the effects of legislation that is known to be effective for specific populations or contexts. For instance, child-specific interventions like pool fencing or supervisory requirements may have measurable effects on child drowning but these would be diluted in an all-age national rate. Similarly, occupational drowning regulations targeting the fishing or maritime sectors may be effective within those populations but invisible at the level of national aggregates. This is a further reason to interpret null findings cautiously and to prioritise disaggregated data in future research. Third, power was a constraint when it came to the regional data. The regional dataset covered just 19 countries, yielding wide confidence intervals and unstable estimates, particularly for interactions. Finally, we also cannot rule out the (largely) null findings being related to the methods and/or analytic strategy we employed. These models should be read as exploratory and supportive only. More generally, while we probed robustness and diagnostics, residual confounding by unmeasured factors (like swimming and CPR skills or emergency services capacity for example) cannot be ruled out. Taken together, these limitations suggest caution in interpreting null legal effects at the national level and point to the need for longitudinal, disaggregated data on both drowning outcomes and the content, enforcement, and uptake of laws.

## Conclusions

This paper provides the first global analysis of drowning prevention legislation and its association with mortality rates. While there are a number of examples that show that regulation has a role in reducing drowning, our findings underscore that these effects do not readily translate at the global level. Across 127 countries, structural determinants, income, urbanisation, alcohol use, disaster exposure, and access to safe water and sanitation, were consistently more predictive of drowning mortality than the presence or number of legislative measures. In most cases, laws themselves showed no systematic association with drowning rates, except for tentative evidence that fencing requirements for public pools and alcohol restrictions near water may reduce deaths when paired with strong institutional capacity and enforcement. The implications are twofold. First, legislation alone is unlikely to be sufficient. Legal instruments must be embedded in wider systems of governance, infrastructure, and social development if they are to deliver meaningful reductions in drowning. Second, our results highlight the pressing need for more granular, disaggregated data, notably in relation to who drowns, under what circumstances, and on the scope, coverage, and enforcement of laws. Without this, efforts to evaluate and improve legal approaches will remain constrained. Ultimately, drowning prevention cannot be advanced by legislation in isolation. Laws matter, but their effectiveness depends on the broader political, social, and environmental context in which they are introduced. Future research should move beyond documenting legal presence to interrogating what kinds of regulatory approaches work, for whom, and under what conditions.

## Supporting information

**S1 Table. Variables and data sources.**
(DOCX)

**S2 Table. Outcome summaries by model.**
(DOCX)

**S3 Table. Countries included in main models.**
(DOCX)

**S4 Table. Regions represented in main models.**
(DOCX)

**S5 Table. Summary of regional models.**
(DOCX)

**S6 Table. Robustness.** NB2 vs PPML.
(DOCX)

**S7 Table. Robustness.** Influence-trimmed NB2 (drop top 5% Cook's D).
(DOCX)

**S8 Table. Countries excluded from influence trimmed NB2.**
(DOCX)

**S9 Table. Dimension reduction.**
(DOCX)

**S10 Table. Diagnostics.**
(DOCX)

**S11 Table. Random forest permutation importance.**
(DOCX)

**S12 Table. Interaction between enforcement capacity × law.**
(DOCX)

**S1 Fig. Summary of regional models.**
(DOCX)

**S2 Fig. Casual DAG: Drowning legislation and morality.**
(DOCX)

## Author contributions

**Conceptualization:** Ryan Essex, Jagnoor Jagnoor.

**Formal analysis:** Ryan Essex.

**Methodology:** Ryan Essex.

**Visualization:** Ryan Essex.

**Writing – original draft:** Ryan Essex, Jagnoor Jagnoor.

**Writing – review & editing:** Ryan Essex, Jagnoor Jagnoor.

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
