## [Decision Letter · Decision Letter 0]

16 Sep 2025

PGPH-D-25-02400

An analysis of global legislation and regulation related to drowning prevention

Dear Dr. Essex,

Thank you for submitting your manuscript to PLOS Global Public Health. After careful consideration, we feel that it has merit but does not fully meet PLOS Global Public Health’s publication criteria as it currently stands. Therefore, we invite you to submit a revised version of the manuscript that addresses the points raised during the review process.

EDITOR: This paper is  very detailed, important, and novel use of global data to analyse drowning mortality and its association with legislation. The contribution is significant, as there has been little comparable work at this scale. That said, a few areas could be clarified and strengthened. The methods would benefit from greater detail on the specific countries included and on any exclusions made, so that readers have a clearer sense of the study population. The list of covariates is extensive, but at times gives the impression that most available variables were included rather than only those with strong evidence for relevance; more justification for their selection would be helpful, and the appendix should ideally avoid listing variables that were not ultimately analysed. The methods and results, while rigorous, could also be written in a way that is slightly easier to follow. Finally, while I recognize the constraints of global data availability, some important factors such as swimming and CPR skills, emergency rescue services, and early warning systems are missing from the analysis. Early warning system coverage in particular may be available through other global datasets, and could be considered for inclusion. At minimum, the absence of such variables should be acknowledged more explicitly.

We look forward to receiving your revised manuscript.

Kind regards,

Uzma Rahim Khan

Academic Editor

Journal Requirements:

1. Please note that PLOS Global Public Health has specific guidelines on code sharing for submissions in which author-generated code underpins the findings in the manuscript. In these cases, all author-generated code must be made available without restrictions upon publication of the work. Please review our guidelines at https://journals.plos.org/globalpublichealth/s/materials-and-software-sharing#loc-sharing-code and ensure that your code is shared in a way that follows best practice and facilitates reproducibility and reuse.

2. Please upload separate figure files in .tif or .eps format. Also, remove the figures from your manuscript file but keep the legends.

4. Please note that your Data Availability Statement is currently missing the repository name and/or the DOI/accession number of each dataset OR a direct link to access each database. If your manuscript is accepted for publication, you will be asked to provide these details on a very short timeline. We therefore suggest that you provide this information now, though we will not hold up the peer review process if you are unable.

Additional Editor Comments (if provided):

Reviewer #1:

Reviewer #2:

Reviewers' comments:

Reviewer's Responses to Questions

**Comments to the Author**

1. Does this manuscript meet PLOS Global Public Health’s publication criteria ? Is the manuscript technically sound, and do the data support the conclusions? The manuscript must describe methodologically and ethically rigorous research with conclusions that are appropriately drawn based on the data presented.

Reviewer #1: Yes

Reviewer #2: No

2. Has the statistical analysis been performed appropriately and rigorously?

Reviewer #1: Yes

Reviewer #2: Yes

3. Have the authors made all data underlying the findings in their manuscript fully available (please refer to the Data Availability Statement at the start of the manuscript PDF file)?

Reviewer #1: Yes

Reviewer #2: No

4. Is the manuscript presented in an intelligible fashion and written in standard English?

Reviewer #1: Yes

Reviewer #2: No

Reviewer #1: This is an ambitious and timely article assessing the association between drowning-related legislation and mortality across countries. The topic is of clear global health relevance, and the authors’ effort to compile a cross-national dataset and triangulate multiple modelling approaches is commendable. The paper is well-written, transparent about methods, and presents extensive supplementary analyses. However, there are important methodological and interpretive issues that weaken the strength of the conclusions. In particular, sample size limitations, potential overfitting, choice of outcome modelling, and lack of clarity around causal interpretation require careful attention before publication.

Abstract

• Strengths: Clear statement of objectives and methods; multiple models (OLS, GLM, RF) reported; key findings succinctly conveyed.

• Concerns:

o Wording overstates null results (e.g., “laws do not matter”); should instead emphasize “no consistent association observed at country level.”

o Statistical detail is limited include effect sizes and 95% CIs for at least one key predictor to give context.

o Policy relevance is implied but not explicitly linked to the observed dominance of structural determinants.

o if Journal permits then add key words from MESH

Introduction

• Strengths: Good framing of drowning as a global public health issue and rationale for examining legislation.

• Concerns:

o Literature review is skewed towards descriptive burden; limited discussion of prior evaluations of policy effectiveness (e.g., natural experiments, case studies).

o Causal pathways (how laws may reduce drowning through enforcement, infrastructure, or behaviour change) need clearer articulation.

o The introduction should foreshadow limitations of ecological cross-country comparisons (endogeneity, measurement error).

Methods

• Strengths: Transparent description of data sources; multiple model specifications attempted; inclusion of robustness checks.

• Concerns:

1. Outcome modelling: OLS on rates is suboptimal; drowning deaths are count data → negative binomial or Poisson with population offset would be more appropriate. Gamma GLM is not well justified.

2. Covariate selection: Many structural predictors included simultaneously despite ~100 observations; risks overfitting. Consider penalized regression or dimension reduction (e.g., PCA).

3. Regional models (n=19): Severely underpowered for multiple predictors and interactions; these should be presented as exploratory only.

4. Law measurement: Binary presence/absence recoding risks attenuation bias; more nuanced scoring of comprehensiveness or enforcement would be preferable.

5. Causality: Cross-sectional design cannot address temporality; countries may adopt laws in response to drowning burden. This limitation should be made explicit.

6. Diagnostics: No evidence provided of checking residuals, multicollinearity (VIFs), or spatial autocorrelation. These are essential for ecological analyses.

Results

• Strengths: Clearly presented tables; coefficients and CIs reported; random forest adds triangulation.

• Concerns:

1. Interpretation: Emphasis on null law effects is too strong given wide CIs and measurement limitations.

2. Sample size: Regional models produce negative adjusted R²; interpret cautiously or drop from main text.

3. Robustness checks: Outlier exclusion changes significance of key covariates, suggesting instability. These shifts should be highlighted.

4. Consistency: Urbanisation, alcohol, and sanitation repeatedly emerge as significant predictors, this stability deserves more emphasis.

5. Interactions: Law × enforcement interactions have huge SEs and are uninterpretable at n=19; presenting them risks misleading readers.

6. Models 4-6 (n=19): The results of these models should be interpreted with extreme caution. With 10+ parameters and n=19, the models are severely overfit (as indicated by negative adjusted R² values). The coefficients are unstable and unreliable. These results should be de-emphasized in the main text and moved to the supplement with a strong caveat.

Discussion

• Strengths: Honest acknowledgment of some limitations; appropriate recognition of structural determinants.

• Concerns:

1. Overgeneralisation — concluding that “laws do not reduce drowning” is not supported; better phrasing: “we did not detect systematic associations at country level.”

2. Limited comparison to case-based or quasi-experimental evidence showing effectiveness of pool fencing or boating regulations.

3. Policy recommendations are vague; should connect observed results (structural determinants dominate) to integrated strategies combining legislation with infrastructure and enforcement.

4. Small sample and measurement issues are mentioned but downplayed; they deserve central placement.

5. Reframing Null Results: The authors appropriately avoid overstating the null findings, correctly pointing to data limitations and the potential for legislation to be effective in specific contexts. They could strengthen this by more explicitly stating that the study finds no evidence of an association at the ecological level, rather than stating legislation is ineffective. The distinction is important.

6. Limitations: The limitations section is good but should be more forceful on two points: 1) The ecological fallacy should be named and discussed as a primary limitation. 2) The severe power issue with the regional models (n=19) must be highlighted as a major constraint.

Tables and Figures

• Strengths: Informative and mostly well formatted.

• Concerns:

o Forest plots or standardized coefficients would aid comparability of effect sizes.

o Tables with >10 predictors for n=19 regions are not meaningful; consider moving to appendix.

o Random forest importance plots should include uncertainty estimates (bootstrapped importance).

Supplementary Material

• Strengths: Very transparent, showing multiple specifications, data sources, and robustness checks.

• Concerns:

o Gamma GLM results difficult to interpret; justify inclusion or simplify.

o Outlier-exclusion results should specify which countries were dropped.

o Regional analyses should be explicitly marked exploratory.

The article addresses a relevant question and shows commendable transparency. However, methodological and interpretive revisions are required:

1. Reanalyse with count-based models (negative binomial with population offset).

2. Downplay or remove unstable regional/interaction models.

3. Clarify measurement limitations of law variables.

4. Reframe conclusions to avoid overstatement and highlight structural determinants.

5. Expand on diagnostics (collinearity, spatial autocorrelation, influence analysis).

Reviewer #2: This study analyzes global legislation and regulation related to drowning prevention.

Please see my comments below.

Abstract

A scientific study should be replicable. In the abstract, the study type is unclear. Insufficient methodological details are given to the reader to understand what and how it was done?

Introduction

The introduction is written in an op-ed style. Most statements are without references. The literature search and knowledge gap are not clearly defined.

Novelty

This data is utilized from WHO report. What new information is it adding to the literature?

Methods

Insufficient details regarding the 127 countries so we cannot understand who is included in this analysis. Grouping and interpreting high-Income countries and low-income countries together statistically is not the right approach to deduce meaningful results.

Results and Discussion

The findings identify known predictors of drowning, with negative results for legislative measures, ultimately not revealing anything meaningful and novel to add to the literature

what does this mean?. If published, this will include your full peer review and any attached files.). If published, this will include your full peer review and any attached files.

**Do you want your identity to be public for this peer review?** For information about this choice, including consent withdrawal, please see our Privacy Policy .

Reviewer #1: **Yes:** Dr.Farhad AliDr.Farhad Ali

Reviewer #2: No

---

## [Decision Letter · Decision Letter 1]

23 Nov 2025

PGPH-D-25-02400R1

An analysis of global legislation and regulation related to drowning prevention

Dear Dr. Essex,

Thank you for submitting your manuscript to PLOS Global Public Health. After careful consideration, we feel that it has merit but does not fully meet PLOS Global Public Health’s publication criteria as it currently stands. Therefore, we invite you to submit a revised version of the manuscript that addresses the points raised during the review process.

We look forward to receiving your revised manuscript.

Kind regards,

Uzma Rahim Khan

Academic Editor

Journal Requirements:

1. Please note that PLOS Global Public Health has specific guidelines on code sharing for submissions in which author-generated code underpins the findings in the manuscript. In these cases, all author-generated code must be made available without restrictions upon publication of the work. Please review our guidelines at https://journals.plos.org/globalpublichealth/s/materials-and-software-sharing#loc-sharing-code and ensure that your code is shared in a way that follows best practice and facilitates reproducibility and reuse.

Additional Editor Comments (if provided):

The authors response has improved the manuscript but there are additional points to be considered.

Reviewers' comments:

Reviewer's Responses to Questions

**Comments to the Author**

Reviewer #1: All comments have been addressed

Reviewer #3: All comments have been addressed

publication criteria ? Is the manuscript technically sound, and do the data support the conclusions? The manuscript must describe methodologically and ethically rigorous research with conclusions that are appropriately drawn based on the data presented.

Reviewer #1: Yes

Reviewer #3: Yes

3. Has the statistical analysis been performed appropriately and rigorously?

Reviewer #1: Yes

Reviewer #3: No

4. Have the authors made all data underlying the findings in their manuscript fully available (please refer to the Data Availability Statement at the start of the manuscript PDF file)?

Reviewer #1: Yes

Reviewer #3: Yes

5. Is the manuscript presented in an intelligible fashion and written in standard English?

Reviewer #1: Yes

Reviewer #3: Yes

Reviewer #1: (No Response)

Reviewer #3: This is an important study with global health implications. The authors used multiple sources of data and with a comprehensive set of structural covariates indicating major socioeconomic, environmental, and institutional determinants of drowning mortality. Also, for exposure, authors considered a variety of measures like the total count, enforcement strength, and their interaction an used multiple modeling strategies to model the relationship with exposure and outcome.

However, these are a few areas especially in design and methodology which if clarified or improved can further strengthen the paper.

Aim 3:"whether strength of enforcement and institutional capacity were influential when it came to the effectiveness of legislation/regulation." Can authors possibly rephrase it in scientific and quantifiable language?

Measurement of Variables: Can authors elaborate how did they measure the extent to which legislative measures were enforced? A clear definition of the enforcement concept is crucial for interpreting the results. Further, to enhance transparency, please provide a detailed list of the nine covariates included, categorized by their domains (e.g., economic, environmental). Additionally, please provide an expanded explanation of the scoring mechanism used to derive the total number of legislative instruments (range 0-19) and the total enforcement score (range 0-190). If possible, can authors include a table of Key explanatory variables, their data source, original form, transformation, and scoring. Can authors spell time the abbreviation the first time they are used such as GDP PPP. Also, how legislative transparency and enforcement translates into V-DEM?

Reference and Citations: Please ensure full citations and/or direct links are provided for all external datasets, including the Varieties of Democracy (V-DEM), World Justice Project (WJP), Global Health Security (GHS), and the World Bank’s World Data Indicators (WDI). Additionally, please provide specific references/citations to support the statement that the control variables were selected based on existing theoretical and empirical evidence on drowning prevention.

Analytic Strategy: Please elaborate on the specific link function used for the outcome variable, which is the drowning mortality rate per 100,000 population. Although OLS was used, which assumes a linear outcome, given that the outcome is a rate derived from count data, please justify the OLS choice over alternative models like Poisson or Negative Binomial regression. Furthermore, please detail the process for checking OLS assumptions (linearity, normality of residuals, etc.) and discuss the theoretical or empirical sources of heteroskedasticity that necessitated the use of robust variance estimators.

The stated goal 'to estimate the impact of legislation' implies a causal objective and thus the methods and language should be consistently aligned. For example, if your goal was to estimate the impact of legislation on national drowning mortality, then legislation is your main primary exposure and has to be included in all nested models. I suggest authors read on Causal Inference in Epidemiology. See this for example: Daniel Westreich, Sander Greenland, The Table 2 Fallacy: Presenting and Interpreting Confounder and Modifier Coefficients, American Journal of Epidemiology, Volume 177, Issue 4, 15 February 2013, Pages 292–298, https://doi.org/10.1093/aje/kws412 and this: https://phlr.temple.edu/sites/phlr/files/documents/CPHLR-TheoryMethods2023_NaturalExperiments.pdf.

I see that authors have multiple aims and that authors used suite of different traditional regression and machine learning analysis. Can authors identify which analysis addressed which objective? Like Exploratory machine-learning classification of drowning risk addressed which objective? How aim 2 and 3 were addressed? Also reconsider model specification again and if possible make a DAG to examine which variable are confounders (so called controls), mediator or collider? Because, if the included covariates (like GDP and urbanization) are colliders or mediators (depending on the specific causal pathway) introducing them could introduce bias or even mask exposure outcome relationship.

Results: Consider streamlining the presentation of results to focus on those that directly address the stated objectives. If authors think information in figure 1 and table 1 is overlapping, then chose to keep anyone of these in main paper and rest can go in supplementary information. Second, for the interpretation of the main model, refer to paper by Westreich referred above, only interpretating the main effect. Third, to provide essential context for the cross-national comparison, please include a table listing the included countries/geographies, alongside two key descriptive features: (1) a measure of inherent risk (e.g., proximity to water bodies, coastline length) and (2) their baseline legislative coverage (e.g., total count/score)

Interpretation/Discussion: Can authors address the possibility that the null finding of no effect of legislation on drowning could be an artifact due to methodology (design and analytic strategy) rather than conclusive evidence.

what does this mean?. If published, this will include your full peer review and any attached files.). If published, this will include your full peer review and any attached files.

**Do you want your identity to be public for this peer review?** For information about this choice, including consent withdrawal, please see our Privacy Policy .

Reviewer #1: No

Reviewer #3: **Yes:** Salima KeraiSalima Kerai

---

## [Decision Letter · Decision Letter 2]

12 Feb 2026

PGPH-D-25-02400R2

An analysis of global legislation and regulation related to drowning prevention

Dear Dr. Ryan Essex,

Thank you for submitting your manuscript to PLOS Global Public Health. After careful consideration, we feel that it has merit but does not fully meet PLOS Global Public Health’s publication criteria as it currently stands. Therefore, we invite you to submit a revised version of the manuscript that addresses the points raised during the review process.

We look forward to receiving your revised manuscript.

Kind regards,

Alex Joseph

Academic Editor

Journal Requirements:

Additional Editor Comments (if provided):

Reviewers' comments:

Reviewer's Responses to Questions

**Comments to the Author**

Reviewer #3: All comments have been addressed

Reviewer #4: (No Response)

publication criteria ? Is the manuscript technically sound, and do the data support the conclusions? The manuscript must describe methodologically and ethically rigorous research with conclusions that are appropriately drawn based on the data presented.

Reviewer #3: Yes

Reviewer #4: Yes

3. Has the statistical analysis been performed appropriately and rigorously?

Reviewer #3: Yes

Reviewer #4: (No Response)

4. Have the authors made all data underlying the findings in their manuscript fully available (please refer to the Data Availability Statement at the start of the manuscript PDF file)?

Reviewer #3: Yes

Reviewer #4: Yes

5. Is the manuscript presented in an intelligible fashion and written in standard English?

Reviewer #3: Yes

Reviewer #4: Yes

Reviewer #3: (No Response)

Reviewer #4: Great revisions so far! Almost ready for publication.

Please see below some minor revision suggestions:

-Clarify what kind of “null” this is (statistical vs. policy-relevant), why this matters, this paper is careful, but readers may still misread “no association” as “laws don’t matter." Authors can consider adding 2–3 sentences distinguishing statistical null findings at the ecological level from policy irrelevance (particularly emphasizing power, heterogeneity, and aggregation bias).

-To improve policy translation, the authors might briefly illustrate their findings with a concrete example (e.g., pool fencing in high-capacity vs. low-capacity settings, or alcohol regulation near water in differing enforcement contexts)..no new data required. Just narrative synthesis.

-Recommend authors adding a brief clarification that permutation importance reflects predictive contribution rather than causal influence, to prevent misinterpretation by non-technical readers.

-Consider highlighting earlier in the methods section that death counts were reconstructed from rates and population, and briefly noting how rounding error was handled (for method transparency and replication of the study).

-For data limitations, the authors could add few lines on how national-level aggregation may mask child-specific or occupational drowning effects where legislation is known to be effective..

what does this mean?. If published, this will include your full peer review and any attached files.). If published, this will include your full peer review and any attached files.

**Do you want your identity to be public for this peer review?** For information about this choice, including consent withdrawal, please see our Privacy Policy .

Reviewer #3: No

Reviewer #4: **Yes:** Khushbu BalsaraKhushbu Balsara

---

## [Editor Report · Decision Letter 3]

11 Mar 2026

An analysis of global legislation and regulation related to drowning prevention

PGPH-D-25-02400R3

Dear Ryan,

We are pleased to inform you that your manuscript 'An analysis of global legislation and regulation related to drowning prevention' has been provisionally accepted for publication in PLOS Global Public Health.

Best regards,

Alex Joseph

Academic Editor